# Coronary Slow-Flow Phenomenon in Takotsubo Syndrome: The Prevalence, Clinical Determinants, and Long-Term Prognostic Impact

**DOI:** 10.3390/ijms25021297

**Published:** 2024-01-20

**Authors:** Konrad Stępień, Karol Nowak, Aleksandra Karcińska, Grzegorz Horosin, Alicia del Carmen Yika, Julia Lenart, Anna Górowska, Sylwia Iwańczyk, Mateusz Podolec, Aleksander Siniarski, Jadwiga Nessler, Jarosław Zalewski

**Affiliations:** 1Department of Coronary Artery Disease and Heart Failure, Jagiellonian University Medical College, 31-202 Krakow, Poland; k.nowak.uj@gmail.com (K.N.); podolecmateusz@gmail.com (M.P.); aleksandersiniarski@gmail.com (A.S.); jnessler@interia.pl (J.N.); j.zalewski@szpitaljp2.krakow.pl (J.Z.); 2Department of Thromboembolic Disorders, Jagiellonian University Medical College, 31-202 Krakow, Poland; 3Student Research Group at Department of Coronary Artery Disease and Heart Failure, Jagiellonian University Medical College, 31-202 Krakow, Poland; aleksandrakarcinska@gmail.com (A.K.); grzegorz.horosin@gmail.com (G.H.); alicia.yika@student.uj.edu.pl (A.d.C.Y.); julialenart3@gmail.com (J.L.); agorowska07@gmail.com (A.G.); 41st Department of Cardiology, Poznan University of Medical Sciences, 61-848 Poznan, Poland; syl.iwanczyk@gmail.com; 5Center for Innovative Medical Education, Jagiellonian University Medical College, 30-688 Krakow, Poland

**Keywords:** takotsubo syndrome, coronary slow flow, long-term prognosis, endothelium

## Abstract

Patients with takotsubo syndrome (TTS) may present coronary slow flow (CSF) in angiography performed in the acute myocardial infarction (MI). However, the detailed clinical relevance and its long-term impact remain poorly understood. Among 7771 MI patients hospitalized between 2012 and 2019, TTS was identified in 82 (1.1%) subjects. The epicardial blood flow was assessed with thrombolysis in myocardial infarction (TIMI) scale and corrected TIMI frame count (TFC), whereas myocardial perfusion with TIMI myocardial perfusion grade (TMPG). CSF was defined as TIMI-2 or corrected TFC > 27 frames in at least one epicardial vessel. CSF was identified in 33 (40.2%) TTS patients. In the CSF-TTS versus normal-flow-TTS group, lower values of left ventricular ejection fraction on admission (33.5 (25–40) vs. 40 (35–45)%, *p* = 0.019), more frequent midventricular TTS (27.3 vs. 8.2%, *p* = 0.020) and the coexistence of both physical and emotional triggers (9.1 vs. 0%, *p* = 0.032) were noted. Within a median observation of 55 months, higher all-cause mortality was found in CSF-TTS compared with normal-flow TTS (30.3 vs. 10.2%, *p* = 0.024). CSF was identified as an independent predictor of long-term mortality (hazard ratio 10.09, 95% confidence interval 2.12–48.00, *p* = 0.004). CSF identified in two-fifths of TTS patients was associated with unfavorable long-term outcomes.

## 1. Introduction

Takotsubo syndrome (TTS) represents a clinical syndrome characterized by an acute and transient ventricular dysfunction, with regional wall abnormalities extending beyond a single epicardial distribution. Noteworthily, in most cases, there are no significant lesions in the performed coronarography [1]. It has been reported previously that approximately 1–2% of clinically suspected acute coronary syndrome (ACS) cases are diagnosed as TTS [2]. Typically, TTS occurs in adults aged > 50 years, predominantly postmenopausal women, with the vast majority of these patients Caucasian [2,3]. Although, in most cases, the TTS course is self-limiting, it is associated with a substantial risk of mortality and possible complications, such as acute heart failure, acute mitral regurgitation or cardiac rupture [1,4]. The exact TTS etiology remains unclear. A typical clinical presentation indicates a preceding psychological or physical stress-inducing incident. Therefore, the most frequently proposed mechanism is excessive sympathetic stimulation and increased catecholamine release. It leads to numerous cardiovascular effects, such as cardiac stunning, vasospasm and microcirculatory dysfunction, but also subclinical general inflammation [2].

The coronary slow-flow phenomenon (CSF) is defined as the angiographically normal or near-normal coronary arteries with delayed antegrade opacification of the distal vasculature [5]. In 2012, Beltrame proposed specific CSF criteria, including the lack of epicardial lesions and delayed distal vessel contrast opacification, as evidenced by either thrombolysis in myocardial infarction (TIMI) score 2 or corrected TIMI frame count (TFC) >27 frames in at least one epicardial vessel [6]. Although the precise pathophysiological mechanism underlying CSF remains undetermined, microvascular and endothelial dysfunction have been identified as the most plausible causes [7]. Impaired endothelial function, as assessed through flow-mediated dilatation (FMD) of the brachial artery, was associated with CSF [8]. This implies that endothelial dysfunction manifests as a widespread process affecting both coronary and peripheral vasculature [8]. Moreover, patients with CSF exhibit abnormal endothelial function, as assessed by concentrations of endothelin-1 and nitric oxide, along with an anomalous response to exercise when compared to healthy controls [9]. Finally, a higher plasma concentration of asymmetric dimethylarginine was found in CSF patients, supporting the hypothesis of endothelial damage in this patient population [10]. Previous studies demonstrated that patients with TTS may present with CSF in coronary angiography performed in the acute phase [11,12,13]. In a recent study, Montone et al. showed, for the first time, that CSF in patients with TTS is associated with a worse clinical outcome. Patients with CSF compared with normal coronary flow have a worse clinical presentation on admission, more frequent in-hospital complications, and higher all-cause long-term mortality [14]. Nevertheless, the precise clinical significance and its long-term prognostic implications have not been conclusively established.

Therefore, we sought to investigate the prevalence of CSF in TTS, its clinical determinants as well as long-term prognostic impact.

## 2. Results

### 2.1. Clinical Characteristics

Among 82 TTS patients, CSF was identified in 33 patients (CSF-TTS, 40.2%) in angiography in the acute phase of MI (median age: 68 (63–76) years) (Figure 1). Both groups were predominantly composed of women (Table 1). 

There were no differences in cardiovascular risk factors, prior cardiovascular or cerebrovascular events, pre-existing psychiatric disorders as well as drugs used before admission and prescribed on discharge between patients with CSF-TTS and normal-flow TTS (Table 1). No differences were found in laboratory parameters on admission and in peak values of myocardial necrosis markers during hospitalization (Table 2).

On admission, TTS patients with CSF had lower values LVEF (33.5 (25–40) vs. 40 (35–45)%, *p* = 0.019) compared with patients with normal flow. However, there were no differences in values of LVEF during hospitalization and on discharge. There were no differences in clinical presentation, the frequency of ventriculography, Killip class on admission as well as the length of the index hospitalization (4 (2–5) vs. 4 (3–6) days, *p* = 0.19). TTS patients with CSF were characterized by a more frequent midventricular type of TTS (27.3 vs. 8.2%, *p* = 0.020) as well as by more frequent coexistence of both physical and emotional triggers (9.1 vs. 0%, *p* = 0.032). 

### 2.2. Angiographic Characteristics 

Significant stenoses in epicardial vessels (>50%) were observed in 9.1% of CSF-TTS and 24.5% of normal-flow TTS, respectively (Table 3). 

Nearly normal coronary arteries (<30% stenosis) were noted more frequently in the CSF-TTS group (75.8 vs. 49.0%, *p* = 0.015). All CSF-TTS patients (*n* = 33) had a slow-flow phenomenon in the left anterior descending (LAD) artery. Of them, 7 patients (21.2%) had isolated CSF in LAD, whereas 26 (78.8%) revealed CSF also in the left circumflex artery (LCx) and 11 (33.3%) in the right coronary artery (RCA) (Table 3). CSF-TTS patients were characterized by higher TFC in each analyzed coronary vessel, including LAD (47.6 (40.6–54.7) vs. 25.9 (21.2–28.2) frames, *p* < 0.001), LCx 54 (39–72) vs. 33 (24–42) frames, *p* < 0.001) and RCA (27 (18–36) vs. 20 (12–21) frames, *p* = 0.008) (Figure 2).

Moreover, patients with CSF exhibited diminished values in myocardial perfusion. Namely, TMPG-3 was reported less frequently in the CSF-TTS group in each analyzed artery, including LAD (33.3 vs. 93.9%, *p* < 0.001), LCx (33.3 vs. 91.8%, *p* < 0.001) and RCA (30.3 vs. 87.8%, *p* < 0.001) (Table 3).

### 2.3. Long-Term Mortality and Its Determinants 

Over a median observation of 55 (28.2–70.1) months, all-cause mortality was higher in the CSF-TTS group than in normal-flow TTS (30.3 vs. 10.2%, *p* = 0.024) (Figure 3). 

The Cox proportional regression analysis showed that the presence of CSF expressed as TIMI-2 flow (*p* = 0.017) independently increased the risk of long-term mortality (Table 4). In turn, higher BMI (*p* = 0.040) and LVEF on admission (*p* = 0.008) were associated with improved long-term survival (Table 4).

## 3. Discussion

In the current study, based on the tertiary single-center registry, we demonstrated that the CSF phenomenon in the coronary angiography performed on admission occurs in a substantial percentage of TTS patients, predominantly in the left coronary artery. Furthermore, it is associated with worse myocardial perfusion expressed with TMPG. Following our results, CSF-TTS patients were characterized more often by a lower LVEF on admission, midventricular type of TTS, coexistence of physical and emotional triggers and near-normal coronary arteries with <30% stenosis. In the longest follow-up to date, we demonstrated that the CSF-TTS prognosis is significantly unfavorable. Despite the lack of CSF presence in the current TTS risk scales, CSF together with lower BMI and LVEF on admission are independent predictors of long-term all-cause mortality.

As mentioned above, there are several data on the coexistence of CSF and TTS in the literature [11,12,13,14]. However, most of them are based on small patient series, which made it impossible to determine the real CSF-TTS incidence. Bybee et al. showed that TFC was abnormal in all 16 women with TTS and that CSF was generalized in all coronary arteries [15]. In a study by Kurisu et al., TFC was significantly higher in 28 TTS patients compared to control in all coronary arteries immediately after onset as well as even after the resolution of left ventricular dysfunction [16]. Khalid et al. showed significantly higher TFC only in the left anterior descending artery in 16 TTS patients [12,13]. Moreover, de Caterina et al. indicated that TFC is more impaired in 25 TTS patients than in STEMI with reperfusion but less than in STEMI with microvascular obstruction [13]. The most reliable CSF-TTS prevalence data to date were provided by Montone et al. [14]. As has been established, CSF occurred in 18 (17.8%) TTS patients. The CSF-TTS incidence in our study is, therefore, clearly higher. Moreover, as many as 17 patients had isolated CSF only in the left anterior descending artery [14]. There are several potential explanations for these discrepancies. Firstly, both analyzed populations, despite similar numbers, differed significantly in some clinical features. In our study, we included more STEMI patients with lover LVEF values. In turn, the Killip class was significantly higher in the study by Montone et al. [14], which may suggest differences in the length of the prehospital phase. Secondly, although the criteria for CSF were the same, the differences could also result from divergence in coronarography interpretation and analysis. Finally, this could be a result of differences between Italian and Polish populations. This may be suggested by differences in the occurrence of individual triggers. The lack of an established trigger was clearly more common in our study.

The exact pathophysiology of CSF in TTS remains unclear. Traditionally, the CSF has been linked with excessive sympathetic stimulation and catecholamine release observed in TTS patients, leading to acute microvascular coronary vasoconstriction and myocardial stunning [17]. This would explain the more common coexistence of physical and emotional triggers observed in our study in the CSF group. Moreover, the interesting hypothesis of neurocardiogenic stunning referring to the neurons that originate in the brainstem and mediate vasoconstriction has been recently promoted [18]. The first reports on invasive functional testing currently recommended for the assessment of microcirculation indicate that coronary microvascular dysfunction is common in patients with TTS and is even more frequent than in patients with ischemia with nonobstructive coronary arteries (INOCA) [19]. Noteworthily, in our study, we did not state a higher incidence of atrial fibrillation and more advanced stenoses in coronary arteries in the CSF group, which substantially reduces the risk of false-positive results [20]. Another possible mechanism explaining CSF is deteriorated endothelial function. This could be evaluated by FMD of the brachial artery, which was previously linked to CSF [8]. Additionally, individuals with CSF display endothelial dysfunction, as determined by biomarker assessment, namely endothelin-1 and nitric oxide concentrations or plasma levels of asymmetric dimethylarginine [8,9]. The endothelial dysfunction markers could be indirectly associated with those observed in this study, myocardial/microvascular perfusion limitation, as demonstrated by TMPG [21,22].

We demonstrated that CSF-TTS patients presented lower values of LVEF on admission compared with normal flow. This finding is consistent with the results by Montone et al. [14], who also showed that patients with TTS and CSF had lower LVEF values on admission (37.3 ± 8.8 vs. 42.8 ± 12.2%, *p* = 0.038). However, we did not observe significant differences in LVEF on discharge, as well as in its improvement during hospitalization. Moreover, ventriculography, an important diagnostic method in TTS, was performed in over half of the patients, without a significant difference between the analyzed groups. The results of ventriculography were consistent with echocardiography in all these patients. Patel et al. showed that the systolic dysfunction of the LV posterolateral segment most accurately distinguishes TTS from STEMI with patent infarct-related artery [23]. As has been well demonstrated in the literature, lower baseline LVEF, especially LVEF < 38%, is an unfavorable prognostic factor [24]. The results, therefore, suggest that in the CSF-TTS group, the microvascular coronary vasoconstriction and myocardial stunning on admission are particularly marked.

The most important conclusion from our study is the negative prognostic role of CSF in long-term TTS follow-up. Our results support the Montone et al. study [14], in which they demonstrated for the first time that the CSF-TTS phenomenon is associated with worse clinical outcomes. During index hospitalization, acute heart failure was more common in patients with CSF (44.4 vs. 18.5%, *p* = 0.02) [9]. In turn, in the follow-up, CSF patients had significantly higher all-cause mortality (50% vs. 22.9%, *p* = 0.011) [14]. Importantly, the follow-up time in the Montone et al. study was 22.6 ± 17.5 months. Therefore, it was more than twice as short as in our study. CSF-TTS was a strong negative predictor of long-term mortality in both studies. In our study, we showed that higher mortality is also connected with lower BMI and LVEF on admission. In turn, Montone et al. [14] associated worse survival with diabetes mellitus, physical triggers, neurological disorders, Killip class III/IV and higher GRACE risk score in addition to CSF. Interestingly, the higher Killip class has a significant negative prognostic impact on long-term outcomes also in a whole population with myocardial infarction with non-obstructive coronary arteries (MINOCA) [25].

Noteworthily, due to the registry data, we focused only on overall mortality. The Montone et al. [14] study demonstrated that mortality in CSF-TTS patients was mostly driven by non-cardiac causes. The results of both studies clearly indicate that CSF in TTS is a strong negative prognostic factor, which is absent in the most commonly used TTS prognostic scales, such as the InterTAK Prognostic Score [26]. The continued prevalence of coronary angiography in the acute phase of TTS makes this indicator easy to obtain and interpret. However, this requires further research on CSF-TTS and, after collecting an appropriate amount of data, the potential validation of a novel prognostic score.

Our study has several limitations. First, this is a single-center registry, and we included a relatively small group of TTS patients. Larger studies may still be needed to confirm our results. Second, only those patients who underwent coronary angiography procedures were eligible for the study, which could lead to selection bias. Third, microvascular perfusion was evaluated by coronary angiography only. We did not perform cardiac magnetic resonance or invasive assessment of microvascular function. However, TFC and TMPG are well-validated and approved methods in the setting of TTS [27,28,29]. Moreover, we did not perform angiographic follow-up to assess the persistence of stated perfusion abnormalities. Finally, because of registry data limitations, we could not analyze the causes of death of these patients.

## 4. Materials and Methods

We conducted a retrospective observational study of patients with myocardial infarction (MI) hospitalized in our tertiary center from 2012 to 2019. Patients with ST-segment elevation of at least 1 mm in at least two contiguous leads were classified as ST-segment elevation MI (STEMI), whereas patients without ST-segment elevation on admission were diagnosed as non-ST-segment elevation MI (NSTEMI) [30,31,32,33]. Among 7771 MI patients, we identified 82 (1.1%) TTS patients in accordance with the InterTAK criteria with performed coronary angiography, and we included them in the analyses (Figure 1) [34]. 

The data, including patients’ demography, anthropometric measurements, cardiovascular risk factors, comorbidities, and concomitant medications, were collected. Furthermore, the laboratory parameters upon admission were systematically collected. Renal failure was diagnosed when the glomerular filtration rate (GFR) calculated using the Cockcroft–Gault formula was lower than 60 mL/min. Anemia was recognized if the hemoglobin level was <13 g/dL for men and <12 g/dL for women. The cut-off value for the thrombocytopenia was 100 × 10^3^/μL [35,36]. Cardiac necrotic biomarkers including isoenzyme MB of creatine kinase (IU/L, upper limit of normal of 24 IU/L) and the high-sensitive cardiac troponin T (ng/mL, upper limit of normal: 0.014 ng/mL) were measured on admission and at least once within the first 24 h. The peak values during the hospitalization were also analyzed. 

The long-term follow-up of all-cause mortality was obtained from the Polish National Death Registry. The study protocol complied with the Declaration of Helsinki and was approved by the Jagiellonian University Medical College Ethics Committee (Consent No. 1072.6120.128.2023; date: 22 November 2023). All included patients gave informed consent.

### 4.1. Echocardiography

Two-dimensional transthoracic echocardiography was performed by a trained physician between the second and fourth day of hospitalization. It was performed at rest in a left decubitus position, using a Vivid S5 ultrasound (GE, Solingen, Germany) equipped with a multi-frequency harmonic transducer, 3Sc-RS (1.3–4 MHz). All measurements were carried out according to the recommendations of the American Society of Echocardiography and the European Association of Cardiovascular Imaging [37]. The left ventricular ejection fraction (LVEF) was assessed on admission as well as sequentially during hospitalization. Based on the obtained echocardiographic image, we divided TTS patients into apical, midventricular, and basal types [34,38].

### 4.2. Angiography and Ventriculography

Coronary angiography was performed within 120 min from diagnosis in patients presenting with STEMI and within 72 h in patients with NSTEMI, in accordance with current guidelines [30]. High-risk NSTEMI patients presenting with haemodynamic instability underwent coronary angiography within 120 min [30].

All coronary angiograms were analyzed offline, using two contralateral projections for each artery at baseline by a cardiologist unaware of the clinical data. According to the guidelines, lesions narrowing the coronary artery by <50% were defined as non-obstructive [36]. All patients with non-obstructive stenosis were divided into two groups with either (i) normal coronary arteries or minimal intracoronary irregularities with stenosis of <30% or (ii) mild to moderate lesions of at least 30 and <50% [36]. Epicardial coronary flow was assessed qualitatively using TIMI flow criteria [27] and quantitatively with TFC [29]. CSF was defined as TIMI-2 or corrected TFC > 27 frames in at least one epicardial vessel [6]. Moreover, myocardial perfusion was assessed with TIMI myocardial perfusion grade (TMPG) [29]. Based on CSF criteria TTS patients were categorized into slow-flow (*n* = 33) and normal-flow (*n* = 49) groups. All analyses were conducted accordingly. Ventriculography was performed in selected patients with dedicated pigtail catheter in the left anterior oblique projection on discretion of the operator.

### 4.3. Statistical Analysis

Continuous variables were expressed as medians (interquartile range) and categorical variables as numbers (percentage). Continuous variables were first checked for normal distribution using the Shapiro–Wilk test. Afterward, differences in the groups among continuous variables were compared by Student’s *t*-test or U-Mann Whitney test if the distribution was normal or different than normal, respectively. Categorical variables were analyzed with the chi-square test or Fisher’s exact test. Kaplan–Meier curves for overall mortality were constructed to estimate the survival rates, and a log-rank test with Bonferroni-corrected threshold was performed to assess the differences in survival between the studied groups. Finally, all independent variables were included in the Cox proportional hazard regression model to determine independent predictors of long-term all-cause mortality. A two-sided *p*-value of less than 0.05 was considered statistically significant. All statistical analyses were performed using STATISTICA software Version 13.3 (StatSoft, Krakow, Poland) or IBM SPSS Statistics Version 26.0 (IBM Corp., Armonk, NY, USA).

## 5. Conclusions

A high proportion of TTS patients were characterized with remarkably marked myocardial perfusion abnormalities, especially in the left coronary vessels, in coronary angiography performed on admission. CSF-TTS patients are characterized by a set of specific clinical features, such as lower LVEF on admission, midventricular type of TTS, the coexistence of physical and emotional triggers and near-normal coronary arteries with <30% stenosis. As has been shown in the longest follow-up so far, CSF was associated with an unfavorable long-term prognosis.

## Figures and Tables

**Figure 1 ijms-25-01297-f001:**
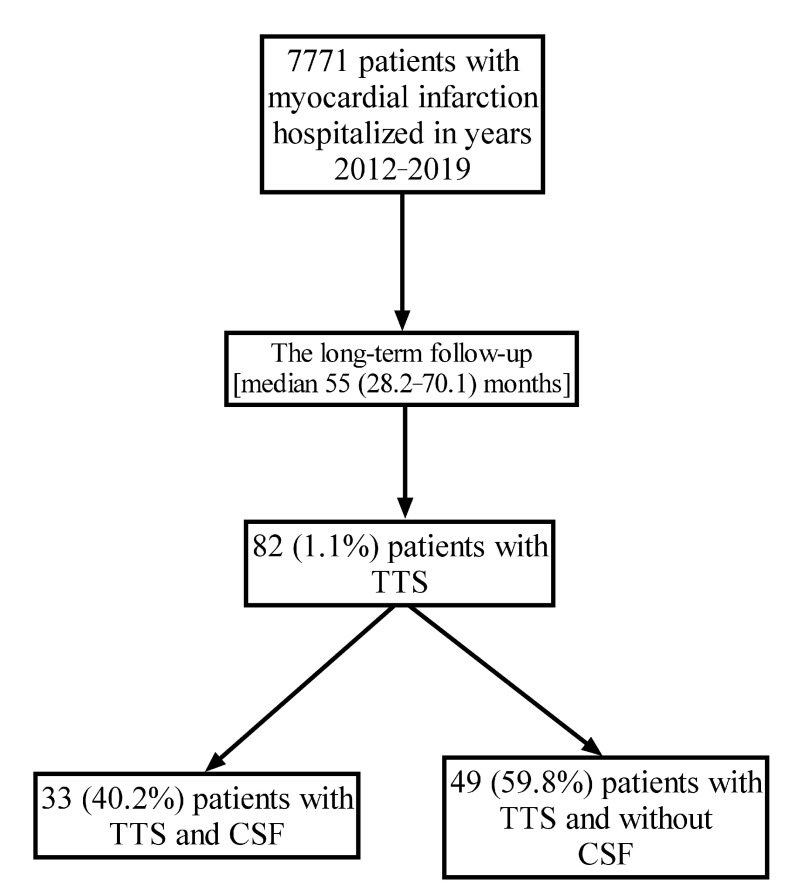
The study flowchart. Abbreviations: TTS, takotsubo syndrome; CSF, coronary slow flow.

**Figure 2 ijms-25-01297-f002:**
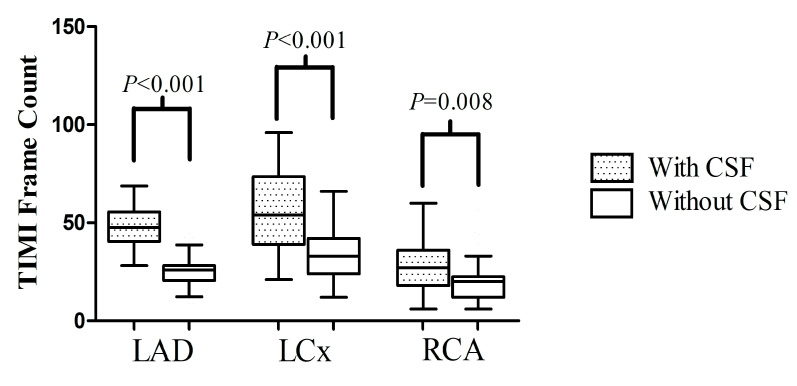
The TIMI Frame Count distribution in the particular coronary arteries. Abbreviations: LAD, left anterior descending artery; LCx, left circumflex artery; RCA, right coronary artery.

**Figure 3 ijms-25-01297-f003:**
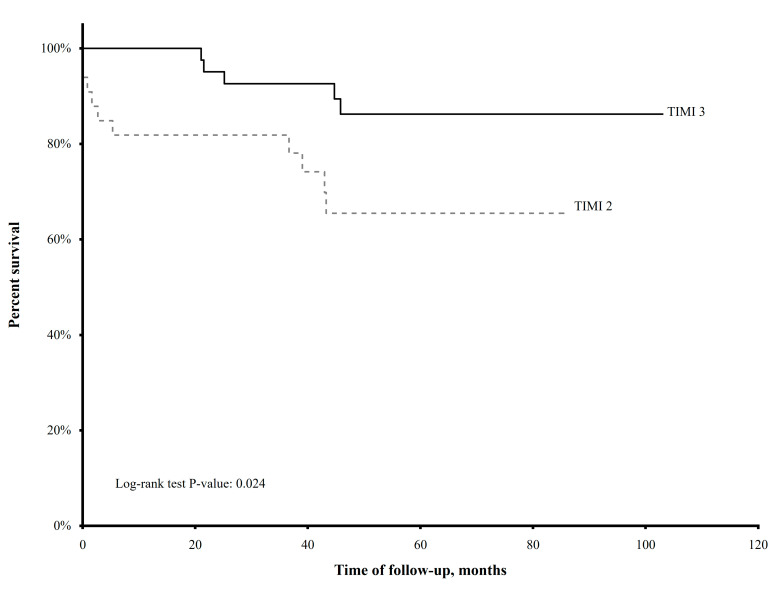
The long-term mortality in the analyzed groups. Abbreviations: TIMI, Thrombolysis in Myocardial Infarction.

**Table 1 ijms-25-01297-t001:** Clinical characteristics of TTS patients with slow and normal coronary flow.

	Slow-Flow TTS *n* = 33	Normal-Flow TTS *n* = 49	*p*-Value
Female gender, %	31 (93.9)	44 (89.8)	0.51
Age, years	68 (63–76)	73 (65–80)	0.56
Body mass index, kg/m^2^	24.2 (23–31.1)	25.7 (23.1–29.7)	0.75
Diabetes mellitus, %	6 (18.2)	11 (22.5)	0.64
Hypertension, %	28 (84.9)	38 (77.6)	0.41
Dyslipidemia, %	20 (60.6)	31 (63.3)	0.81
Renal failure, %	8 (24.2)	17 (34.7)	0.31
Active smoking, %	5 (15.2)	3 (6.1)	0.18
Atrial fibrillation, %	8 (24.2)	6 (12.2)	0.16
Anemia, %	6 (18.2)	5 (10.2)	0.30
Thrombocytopenia, %	1 (3.0)	2 (4.1)	0.80
Prior myocardial infarction, %	2 (6.1)	3 (6.1)	0.99
Prior PCI, %	1 (3.0)	1 (2.0)	0.78
Prior stroke, %	0	1 (2.0)	0.41
Psychiatric disorder, %	3 (9.1)	10 (20.4)	0.17
Chronic obstructive pulmonary disease, %	4 (12.1)	2 (4.1)	0.17
Active or prior malignancy, %	7 (21.2)	4 (8.2)	0.09
Heart failure, %	9 (27.3)	8 (16.3)	0.23
Clinical presentation, %			
NSTEMI	17 (51.5)	23 (46.9)	0.68
STEMI	16 (48.5)	26 (53.1)	
Left ventricular ejection fraction, %			
On admission	33.5 (25–40)	40 (35–45)	0.019
On discharge	45 (35–50)	46 (40–55)	0.13
Ventriculography, %	14 (42.4)	31 (63.3)	0.06
Improvement during hospitalization	12.5 (5–20)	9 (0–15)	0.23
Killip class III/IV on admission, %	4 (12.1)	2 (4.1)	0.17
Length of hospitalization, days	4 (2–5)	4 (3–6)	0.19
Cardiogenic shock, %	4 (12.1)	2 (4.1)	0.17
In-hospital mortality, %	2 (6.1)	0	0.08
Type of TTS, %			
Apical	24 (72.7)	45 (91.8)	0.020
Midventricular	9 (27.3)	4 (8.2)	
Trigger, %			
Physical	7 (21.2)	6 (12.2)	0.28
Emotional	10 (30.3)	18 (36.7)	0.55
Both types	3 (9.1)	0	0.032
Undetermined	19 (57.6)	25 (51.0)	0.56
Drugs before admission, %			
Aspirin	3 (9.1)	2 (4.1)	0.35
P2Y12 inhibitor	0 (0.0)	1 (2.0)	0.41
ACE-I/ARB	21 (63.6)	28 (57.1)	0.56
Beta-adrenolytic	17 (51.5)	17 (34.7)	0.13
Statin	11 (33.3)	12 (24.5)	0.38
Prescribed drugs on discharge, %			
Aspirin	26 (78.8)	40 (81.6)	0.75
P2Y12 inhibitor	10 (30.3)	16 (32.7)	0.82
ACE-I/ARB	30 (90.9)	45 (91.8)	0.88
Beta-adrenolytic	30 (90.9)	44 (89.8)	0.87
Statin	29 (87.9)	40 (81.6)	0.45

Abbreviations: data are shown as median (interquartile range) or number (percentage); TTS, takotsubo syndrome; PCI, percutaneous coronary intervention; NSTEMI, non-ST elevation myocardial infarction; STEMI, ST-elevation myocardial.

**Table 2 ijms-25-01297-t002:** Laboratory parameters in the analyzed groups.

	Slow-Flow TTS *n* = 33	Normal-FLOW TTS *n* = 49	*p*-Value
White blood cells, ×10^3^/µL	11.3 (9–14)	10.4 (8.3–12)	0.19
Hemoglobin, g/dL	13.6 (12.8–14.5)	13.5 (12.6–14.3)	0.59
Platelet count, ×10^3^/µL	263 (221–297)	243 (204–286)	0.24
Creatinine, µmol/L	73 (65–84)	75 (64–89)	0.90
Glomerular filtration rate, ml/min	69.6 (51.2–99.2)	66.2 (50.9–80.5)	0.70
C-reactive protein, mg/L	4 (2–16)	5 (3–12)	0.60
Myocardial necrosis markers—on admission:			
Troponin, ng/ml	0.481 (0.271–0.734)	0.354 (0.196–0.622)	0.23
Creatine kinase, IU/L	184 (117–301)	172 (125–273)	1.00
Creatine kinase MB isoenzyme, IU/L	29 (21–43.5)	27 (19–40)	0.39
Myocardial necrosis markers—peak values:			
Troponin, ng/ml	0.554 (0.290–0.830)	0.413 (0.198–0.785)	0.35
Creatine kinase, IU/L	237 (133–332)	213 (127–339)	0.87
Creatine kinase MB isoenzyme, IU/L	31 (25–47)	31 (19–44)	0.62

Abbreviations: TTS, takotsubo syndrome.

**Table 3 ijms-25-01297-t003:** Angiographic characteristics of the analyzed groups.

	Slow-Flow TTS *n* = 33	Normal-Flow TTS *n* = 49	*p*-Value
>50% stenosis, %	3 (9.1)	12 (24.5)	0.08
30–50% stenosis, %	5 (15.2)	13 (26.5)	0.22
<30% stenosis, %	25 (75.8)	24 (49.0)	0.015
Slow-flow phenomenon, %			
Generalized (LAD + LCx + RCA)	11 (33.3)	-	
Left artery (LAD + LCx)	15 (45.5)	-	
Isolated LAD	7 (21.2)	-	
TFC, frames			
LAD	47.6 (40.6–54.7)	25.9 (21.2–28.2)	<0.001
LCx	54 (39–72)	33 (24–42)	<0.001
RCA	27 (18–36)	20 (12–21)	0.008
TMPG LAD, %			
3	11 (33.3)	46 (93.9)	<0.001
2	15 (45.5)	0	
1	5 (15.2)	2 (4.1)	
0	2 (6.1)	1 (2.0)	
TMPG LCx, %			
3	11 (33.3)	45 (91.8)	<0.001
2	14 (42.4)	1 (2.0)	
1	4 (12.1)	1 (2.0)	
0	4 (12.1)	2 (4.1)	
TMPG RCA, %			
3	10 (30.3)	43 (87.8)	<0.001
2	6 (18.2)	3 (6.1)	
1	13 (39.4)	2 (4.1)	
0	4 (12.1)	1 (2.0)	

Abbreviations: TTS, takotsubo syndrome; LAD, left anterior descending artery; LCx, left circumflex artery; RCA, right coronary artery; TFC, TIMI frame count; TMPG, TIMI myocardial perfusion grade.

**Table 4 ijms-25-01297-t004:** Independent predictors of long-term mortality.

	Univariable				Multivariable			
	*p*-Value	HR	95% CI for HR		*p*-Value	HR	95% CI for HR	
			Lower	Upper			Lower	Upper
Age, per 1 year	<0.001	1.13	1.06	1.20	0.154	1.09	0.97	1.22
BMI, per 1 kg/m^2^	0.008	0.83	0.74	0.95	0.040	0.73	0.54	0.99
LVEF on admission, per 1%	0.003	0.89	0.83	0.96	0.008	0.82	0.70	0.95
TIMI, 2 vs. 3	0.031	3.23	1.11	9.09	0.017	23.81	1.75	81.40
TTS type, 1 vs. 2	0.676	0.73	0.16	3.23	0.071	0.07	0.01	1.27
Creatinine, per 1 µmol/L	<0.001	1.03	1.01	1.05	0.715	0.99	0.96	1.03

Abbreviations: TTS, takotsubo syndrome; TIMI, thrombolysis in myocardial infarction; BMI, body mass index; LVEF, left ventricular ejection fraction; HR, hazard ratio; CI, confidence interval.

## Data Availability

The data presented in this study are available on request from the corresponding author.

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
