# Peer review of "Coronary Slow-Flow Phenomenon in Takotsubo Syndrome: The Prevalence, Clinical Determinants, and Long-Term Prognostic Impact"

_ijms, 2024, doi:10.3390/ijms25021297_

Round 1
Reviewer 1 Report
Comments and Suggestions for Authors
In this article entitled “Coronary slow-flow phenomenon in takotsubo syndrome: the prevalence, clinical determinants, and long-term prognostic impact.” the authors investigate the prevalence of CSF in TTS, its clinical determinants as well as long-term prognostic impact. This is an interesting topic, with possible implications for clinicians.
I think that the article is of potential interest and that it is well written. The figures and tables are clear. Here are my comments:
1. Section “Results” and “Discussion” have been reported before section “Methods”. This is obviously a mistake and therefore the order of the paragraph has to be fixed. As a result, the references order needs to be checked and revised.
2. Lines 232-233: the authors included a reference (number 32) after the following sentence: “The data including patients' demography, anthropometric measurements, cardiovascular risk factors, comorbidities, and concomitant medications were collected”. I don’t think a reference is needed after this sentence (probably the authors wanted to refer to a table). Please remove it and check if it was needed elsewhere.
3. The authors did not mention the ventriculography in the diagnostic methods. As it is one important diagnostic method for TTS, a mention of it should be made. How many patients underwent ventriculography? In how many of them it was consistent with the echographic findings? Please discuss this aspect.
4. In section discussion the authors report that the observed incidence of CSF in TTS in their work is significantly higher than that observed in the most reliable previous study, that of Montone et al. The author should better discuss this aspect and provide a possible explanation for such difference in the observed incidence of CSF.
5. The Data Availability Statement is missing (the one reported is the template from the website).
Author Response
- Section “Results” and “Discussion” have been reported before section “Methods”. This is obviously a mistake and therefore the order of the paragraph has to be fixed. As a result, the references order needs to be checked and revised.
In accordance with the reviewer suggestion we have changed the order of the sections and relocated the "Materials and Methods" section between the "Introduction" and "Results". We have also revised the order of the references.
- Lines 232-233: the authors included a reference (number 32) after the following sentence: “The data including patients' demography, anthropometric measurements, cardiovascular risk factors, comorbidities, and concomitant medications were collected”. I don’t think a reference is needed after this sentence (probably the authors wanted to refer to a table). Please remove it and check if it was needed elsewhere.
The above-mentioned reference was added in the current version of the manuscript after more adequate sentence “Patients with ST-segment elevation of at least 1 mm in at least two contiguous leads were classified as ST-segment elevation MI (STEMI), whereas patients without ST-segment elevation on admission were diagnosed as non-ST-segment elevation MI (NSTEMI)”.
- The authors did not mention the ventriculography in the diagnostic methods. As it is one important diagnostic method for TTS, a mention of it should be made. How many patients underwent ventriculography? In how many of them it was consistent with the echographic findings? Please discuss this aspect.
Thank you for that comment. The ventriculography was performed in 54.9% of patients, without significant difference between analyzed groups (Table 1). The result of ventriculography were consistent with echocardiography in all these patients. We have added appropriate paragraphs in Materials and Methods and Results sections and briefly discussed that issue in the Discussion.
- In section discussion the authors report that the observed incidence of CSF in TTS in their work is significantly higher than that observed in the most reliable previous study, that of Montone et al. The author should better discuss this aspect and provide a possible explanation for such difference in the observed incidence of CSF.
Thank you for that comment. We have proposed several possible explanations of the observed differences in the Discussion section.
- The Data Availability Statement is missing (the one reported is the template from the website).
The Data Availability Statement has been added.
Reviewer 2 Report
Comments and Suggestions for Authors
Stępień et al. wrote an interesting paper regarding the prevalence and long-term prognostic role of coronary slow-flow phenomenon (CSF) in patients with takotsubo syndrome (TTS).
In summary, they found that a significant portion of TTS patients were characterized with CSF on admission, especially in the left coronary vessels. Furthermore, CSF has been shown to be a strong predictor of long-term all-cause mortality in this population.
The main limitations of the study are the small sample size and its single-center design. Despite this, the topic is very interesting and the overall quality of the paper is quite good.
You can find below some points that should be considered to improve the presentation of the results.
- Kindly relocate the "Materials and Methods" section between the "Introduction" and "Results."
- In the "Materials and Methods" section, it would be beneficial to explicitly state that the final population of 82 patients with TTS was categorized into two groups (slow-flow and normal-flow) and that all analyzes were conducted accordingly. Although this division is apparent throughout the paper, explicitly reporting it in the methods section will enhance readability and transparency.
- In the "Angiography" paragraph, please include specific details regarding the timing of coronary angiography in the enrolled patients. It is assumed that the timing corresponds to the type of clinical presentation, with immediate studies for STEMIs and within approximately 72 hours for NSTEMIs, in accordance with the most recent guidelines.
- In the 'Angiography' section, replace the term 'insignificant' with 'non-obstructive.' It's important to note that these two terms are not synonymous, as even plaques causing a stenosis of 20%, though not significant from a hemodynamic perspective, can be clinically relevant.
- Table 1: The authors have not provided information concerning the therapy at the time of hospital admission. Consider filling this gap in your manuscript to offer a more complete view of the treatment context.
- Figure 1 contains an error: it states 79 TTS patients without CSF, but the correct number is 49. Please correct for accuracy."
- In Table 4, consider incorporating LVEF at hospital admission into the multivariable model, especially since it was significantly lower in the slow-flow TTS group.
- In the discussion section, it would be beneficial to highlight that within the context of ACS with unobstructed coronary arteries, the Killip class has proven to be a straightforward and dependable prognostic marker. This holds true not only for patients with TTS but also for those with MINOCA, as corroborated by the findings of this recent study (see PMID: 37596114).
Author Response
- Kindly relocate the "Materials and Methods" section between the "Introduction" and "Results."
In accordance with the reviewer suggestion we have changed the order of the sections and relocated the "Materials and Methods" section between the "Introduction" and "Results". We have also revised the order of the references.
- In the "Materials and Methods" section, it would be beneficial to explicitly state that the final population of 82 patients with TTS was categorized into two groups (slow-flow and normal-flow) and that all analyzes were conducted accordingly. Although this division is apparent throughout the paper, explicitly reporting it in the methods section will enhance readability and transparency.
That information has been additionally added in Materials and Methods section.
- In the "Angiography" paragraph, please include specific details regarding the timing of coronary angiography in the enrolled patients. It is assumed that the timing corresponds to the type of clinical presentation, with immediate studies for STEMIs and within approximately 72 hours for NSTEMIs, in accordance with the most recent guidelines.
We have added that information in the indicated paragraph in Materials and Methods section.
- In the 'Angiography' section, replace the term 'insignificant' with 'non-obstructive.' It's important to note that these two terms are not synonymous, as even plaques causing a stenosis of 20%, though not significant from a hemodynamic perspective, can be clinically relevant.
Thank you for that comment. That correction has been performed.
- Table 1: The authors have not provided information concerning the therapy at the time of hospital admission. Consider filling this gap in your manuscript to offer a more complete view of the treatment context.
The data regarding drugs used before the admission have been placed in Table 1 and commented in the Results section.
- Figure 1 contains an error: it states 79 TTS patients without CSF, but the correct number is 49. Please correct for accuracy.
That mistake in Figure 1 has been corrected.
- In Table 4, consider incorporating LVEF at hospital admission into the multivariable model, especially since it was significantly lower in the slow-flow TTS group.
In agreement with that suggestion we have incorporated LVEF at hospital admission into the multivariable model in the current version of the manuscript (Table 4).
- In the discussion section, it would be beneficial to highlight that within the context of ACS with unobstructed coronary arteries, the Killip class has proven to be a straightforward and dependable prognostic marker. This holds true not only for patients with TTS but also for those with MINOCA, as corroborated by the findings of this recent study (see PMID: 37596114).
Thank you for that comment. The appropriate paragraph commenting that issue has been added in the Discussion section.
Round 2
Reviewer 2 Report
Comments and Suggestions for Authors
Thank you to the authors for the revisions made, which I believe have enhanced the quality of the final manuscript.